# Independent and Joined Association between Socioeconomic Indicators and Pediatric Obesity in Spain: The PASOS Study

**DOI:** 10.3390/nu15081987

**Published:** 2023-04-20

**Authors:** Clara Homs, Paula Berruezo, Albert Arcarons, Julia Wärnberg, Maddi Osés, Marcela González-Gross, Narcis Gusi, Susana Aznar, Elena Marín-Cascales, Miguel Ángel González-Valeiro, Lluis Serra-Majem, Nicolás Terrados, Josep A. Tur, Marta Segú, Montserrat Fitó, Juan Carlos Benavente-Marín, Idoia Labayen, Augusto G. Zapico, Jesús Sánchez-Gómez, Fabio Jiménez-Zazo, Pedro E. Alcaraz, Marta Sevilla-Sanchez, Estefanía Herrera-Ramos, Susana Pulgar-Muñoz, Cristina Bouzas, Raimon Milà, Helmut Schröder, Santiago F. Gómez

**Affiliations:** 1Gasol Foundation Europe, 08830 Sant Boi de Llobregat, Spain; 2Global Research on Wellbeing (GroW), Faculty of Health Sciences, Blanquerna Ramon Llull University, 08025 Barcelona, Spain; 3Office of the High Commissioner against Child Poverty, 28079 Madrid, Spain; 4Department of Sociology, National Distance Education University (UNED), 28012 Madrid, Spain; 5Physiopathology of Obesity and Nutrition Networking Biomedial Research Center (CIBEROBN), Institute of Health Carlos III, 28029 Madrid, Spain; 6EpiPHAAN Research Group, School of Health Sciences, Instituto de Investigación Biomédica en Málaga (IBIMA), University of Málaga, 29590 Málaga, Spain; 7IS-FOOD—Institute for Sustainability & Food Chain Innovation, Universidad Pública de Navarra (UPNA), IDISNA, 31006 Pamplona, Spain; 8ImFINE Research Group, Department of Health and Human Performance, Universidad Politecnica de Madrid, 28003 Madrid, Spain; 9Physical Activity and Quality of Life Research Group (AFYCAV), Faculty of Sport Sciences, University of Extremadura, 10003 Cáceres, Spain; 10PAFS Research Group, Faculty of Sports Sciences, University of Castilla-La Mancha-Toledo Campus, 45071 Toledo, Spain; 11Biomedical Research Networking Center on Frailty and Healthy Aging (CIBERFES), 28029 Madrid, Spain; 12UCAM Research Center for High Performance Sport, UCAM Universidad Católica de Murcia, 30107 Murcia, Spain; 13Facultad de Deporte, UCAM Universidad Católica de Murcia, 30107 Murcia, Spain; 14Strength & Conditioning Society, 30008 Murcia, Spain; 15Faculty of Sports Sciences and Physical Education, Universida de da Coruña, 15001 A Coruña, Spain; 16Preventive Medicine Service, Canarian Health Service, Centro Hospitalario Universitario Insular Materno Infantil (CHUIMI), 35016 Las Palmas de Gran Canaria, Spain; 17Regional Unit of Sports Medicine-Municipal Sports Foundation of Avilés, 33402 Avilés, Spain; 18Research Group of Community Nutrition & Oxidative Stress, University of the Balearic Islands-IUNICS & Health Research Institute of the Balearic Islands (IDISBA), 07122 Palma de Mallorca, Spain; 19Barça Foundation, 08028 Barcelona, Spain; 20Cardiovascular Risk and Nutrition Research Group (CARIN), Hospital del Mar Medical Research Institute (IMIM), 08003 Barcelona, Spain; 21Department of Language, Arts and Physical Education, Universidad Complutense de Madrid, 28040 Madrid, Spain; 22Research Institute of Biomedical and Health Sciences (IUIBS), University of Las Palmas de Gran Canaria, 35016 Las Palmas de Gran Canaria, Spain; 23Health Research Institute of the Principality of Asturias (ISPA), 33011 Oviedo, Spain; 24CIBER Epidemiology and Public Health (CIBERESP), Carlos III Health Institute, 28029 Madrid, Spain; 25Nursing and Physiotherapy Department, University of Lleida, 25198 Lleida, Spain

**Keywords:** obesity, child, adolescent, socioeconomic status, cross-sectional study

## Abstract

Childhood obesity is a public health problem worldwide. An important determinant of child and adolescent obesity is socioeconomic status (SES). However, the magnitude of the impact of different SES indicators on pediatric obesity on the Spanish population scale is unclear. The aim of this study was to assess the association between three SES indicators and obesity in a nationwide, representative sample of Spanish children and adolescents. A total of 2791 boys and girls aged 8 to 16 years old were included. Their weight, height, and waist circumference were measured. SES was assessed using two parent/legal guardian self-reported indicators (educational level -University/non-University- and labor market status -Employed/Unemployed-). As a third SES indicator, the annual mean income per person was obtained from the census section where the participating schools were located (≥12.731€/<12.731€). The prevalence of obesity, severe obesity, and abdominal obesity was 11.5%, 1.4%, and 22.3%, respectively. Logistic regression models showed an inverse association of both education and labor market status with obesity, severe obesity, and abdominal obesity (all *p* < 0.001). Income was also inversely associated with obesity (*p* < 0.01) and abdominal obesity (*p* < 0.001). Finally, the highest composite SES category (University/Employed/≥12.731€ *n* = 517) showed a robust and inverse association with obesity (OR = 0.28; 95% CI: 0.16–0.48), severe obesity (OR = 0.20; 95% CI: 0.05–0.81), and abdominal obesity (OR = 0.36; 95% CI: 0.23–0.54) in comparison with the lowest composite SES category (Less than University/Unemployed/<12.731€; *n* = 164). No significant interaction between composite SES categories and age and gender was found. SES is strongly associated with pediatric obesity in Spain.

## 1. Introduction

Pediatric obesity is a public health problem worldwide [1]. In Spain, the prevalence of excess body weight in children aged 7 to 13 increased from 32.3% (95% CI: 29.1–35.6%) during 1999–2010 to 35.3% (95% CI, 32.9–37.7%) during 2011–2021 [2]. The obesity epidemic had short-, mid-, and long-term consequences on the physical, phycological, and social spheres of health [3]. 

Obesity is the result of complex interactions of multiple determinants, such as nutrition [4], physical activity and sedentary behaviors [5], sleep [6], psychological factors [7], and genetic predisposition [8]. Evidence from high-income countries (HICs) showed an inverted association between socioeconomic status (SES) and pediatric obesity [9,10,11,12]. Nonetheless, a recent review describes the complexity of studying the association between SES and childhood obesity and concludes that more research is needed [11].

A European study showed that in 7 out of the 13 HICs, the prevalence of overweightness was double among children whose parents had a low educational level than among those whose parents had a high educational level [13]. Another study showed that, in the United Kingdom, adolescents from the poorest backgrounds had a two-fold risk of obesity than those from the richest ones [14].

In Spain, the impact of different SES indicators on pediatric obesity in a nationwide, representative sample remains unclear [15] since the evidence is only available for the 6- to 9-year-old population [15]. Three studies that used self-reported anthropometric data from the Spanish National Health Survey (Encuesta Nacional de Salud de España, ENS) that collected information on children aged 0–15 years found that (i) obesity and overweightness showed an inverse association with parental occupation [16]; (ii) there is a growing tendency of obesity and overweightness among adolescents from families with a low educational level [17]; and (iii) there is a higher tendency of obesity and overweightness among adolescents whose parents have a lower economic situation [18]. However, self-reported data tend to overestimate height while underestimating weight [16,17,18]. Moreover, contrasting results were obtained in studies in different regions of Spain. In particular, a longitudinal study from 2006 to 2016 in Catalonia [19] indicated a socioeconomic gradient for childhood obesity: The prevalence was higher both in the most deprived areas and among children with non-Spanish nationalities, especially Africans and Asians. Conversely, a cross-sectional study carried out in 2018 with a representative sample in Andalusia [20] reported that the youngest populated had no significant social gradient for overweightness and obesity prevalence.

The World Health Organization (WHO) highlighted the urgency of reducing socioeconomic inequalities to prevent childhood obesity [21,22]. In Spain, structural measures to tackle childhood obesity that strongly consider health inequalities are being implemented in the framework of the National Plan for the Reduction of Childhood Obesity for 2030 [23] presented in 2022. Other initiatives especially benefitting the young population from low SES backgrounds include applying taxes on sugar-sweetened beverages in Catalonia [24] and regulating the advertising of unhealthy food and beverages to children under the age of 16 [25,26]. Finally, it is crucial to understand the association between SES and weight status among Spanish children and adolescents and to monitor the trend of this association over a prolonged time to apply the appropriate measures to fight childhood obesity. 

The present study aimed to assess the association between three SES indicators (educational level, labor market status, and income) and weight status (obesity, severe obesity, and abdominal obesity) in a nationwide, representative sample of Spanish children and adolescents. We also assessed the joined association between combinations of the three SES indicators and weight status.

## 2. Materials and Methods

### 2.1. Study Design 

This study is a cross-sectional analysis within the framework of the PASOS study (Physical Activity, Sedentarism, and Obesity in Spanish Youth), a nationwide, representative, observational, and multicenter research study. Such project aimed to determine the levels of physical activity, sedentarism, lifestyle factors, and weight status of the Spanish youth population. Details of the PASOS protocol have been fully described [27]. A STROBE list is attached (see Appendix A).

### 2.2. Participants and Recruitment

The participants of the PASOS study were children and adolescents aged 8 to 16 enrolled through randomly selected primary and secondary schools. Data were collected from March 2019 to February 2020 from 244 primary and secondary schools across the 17 Spanish autonomous communities. Informed consent was obtained for each participant, reaching a total of 4025 parents or legal guardians. Data on children’s and adolescent’s lifestyles were collected with self-reported online questionnaires with the support of trained professionals in each school. Additionally, anthropometric measurements (body weight, height, and waist circumference) of the participants were taken by trained professionals during school time.

### 2.3. Anthropometric Variables

Anthropometric variables were taken according to the WHO standardized protocol [28]. Body weight, height, and waist circumference were gathered using an electronic SECA 899 scale, a portable SECA 217 stadiometer, and a flexible non-stretch SECA 201 metric tape, respectively. The body mass index (BMI) (kg/m^2^) was calculated using weight and height measures according to WHO growth charts [29]. Obesity and severe obesity were defined as “>WHO growth reference median + 2 standard deviation (SD)” and “>WHO growth reference median + 3 SD”, respectively [29]. The waist-to-height ratio (WHtR) was calculated by dividing waist circumferences (cm) by height (cm), and abdominal obesity was defined as WHtR ≥ 0.5 [30]. 

### 2.4. Assessment of Socioeconomic Status 

Three SES indicators were gathered: The educational level and labor market status of parents or legal guardians were obtained through a self-reported paper questionnaire; the income was obtained through the database of the Spanish National Statistics Institute on the annual mean income per person in 2019 [31]. 

Each participant’s family received two copies of the adult questionnaire. To Designated Adult 1, parents or legal guardians were asked who spent more time with the child or adolescent. If a second parent or legal guardian answered the second copy of the questionnaire, they were designated as Adult 2. Data from mothers/female legal guardians designated as Adult 1 were prioritized (76.4%) followed by data from mothers/female legal guardians designated as Adult 2 (17.1%) if the parent/legal guardian designated as Adult 1 was male or absent; finally, data from fathers/male legal guardians designated as Adult 1 (6.1%) or Adult 2 (0.4%) were used if data were still lacking.

Parental educational level was categorized into “university” and “less than university”. The latter included general certificate of education, vocational education and training, general certificate of secondary education, primary education, and no education. Parental labor market status was designated as “employed” and “unemployed” excluding other labor market categories, such as household work, student, retirement, and permanent disability. 

The annual mean income per person was assigned to each participant according to the corresponding value for the census section where their school was located. Two categories were defined: (i) above the total sample mean value for this variable (≥12.731€) and (ii) below the mean (<12.731€). 

Composite SES categories were created by combining each indicator and obtaining the less favorable category (Less than University/Unemployed/<12.731€; *n* = 164), the most favorable one (University/Employed/≥12.731€; *n* = 517), and the intermediate ones. 

### 2.5. Other Variables 

The data on gender and age were obtained from the informed consent. Child and adolescent adherence to the Mediterranean diet and minutes dedicated to moderated/vigorous physical activity (MVPA) were self-reported by the KidMed index [32] and PAU7-S questionnaires [33], respectively.

### 2.6. Statistical Analysis

Logistic regression analysis was performed to determine the association between each of the 3 SES indicators and the prevalence of obesity, severe obesity, and abdominal obesity. Furthermore, a composite SES category was calculated including educational level (University/Less than University), labor market status (Employed/Unemployed), and annual mean income per person (Above/Below the studied population mean). The final composite variable of 9 categories was used for analysis after excluding 2 categories (University/Unemployed/<12.731€ and University/Unemployed/≥12.731€) with less than 1% of the study population. The 6 composite SES categories were included as independent variables in logistic regression models with 3 anthropometric outcomes (obesity, severe obesity, and abdominal obesity) being the dependent variables. A first model was adjusted for gender and age and a second model also for school, adherence to the Mediterranean diet, and minutes of MVPA. Additionally, we tested also for interaction between indicators of SES with adherence to the Mediterranean diet and MVPA. 

Moreover, the association of the composite SES categories with age and gender was tested. The associations were considered significant if *p* < 0.05. A flowchart of case selection included in this analysis is attached (see Appendix A). The distribution by regions of the final analyzed sample is also declared and compared with the originally expected distribution (see Appendix A). All statistical analyses were performed using SPSS for Windows version 22 (SPSS, Inc., Chicago, IL, USA).

### 2.7. Ethic Aspects 

The PASOS study was performed according to the guidelines of the Declaration of Helsinki and approved by the Ethics Committee of the Fundació Sant Joan de Déu, Barcelona, Spain, on 17 December 2018 and Reference PIC-179-18. The trial was registered in 2019 at the International Standard Randomized Controlled Trial (ISRCT; https://www.isrctn.com/ISRCTN34251612, accessed on 1 March 2023).

## 3. Results

Table 1 shows that most parents had an educational level lower than a university degree (66.0%) and were employed (90.0%). Regarding income, 45.3% of the participants attended schools located in a census section with an annual mean income per person higher or equal to 12.731€. The prevalence of obesity, severe obesity, and abdominal obesity in children and adolescents was 11.5%, 1.4%, and 22.3%, respectively. 

Figure 1 shows the individual association of each of the three SES indicators with the anthropometric outcomes of the participants. Logistic regression analysis adjusted for gender and age revealed an inverse association of parental educational level with the prevalence of obesity, severe obesity, and abdominal obesity of adolescents (Figure 2). Being employed was also inversely associated with the three anthropometric outcomes. The magnitude of these associations was strongest for high educational levels and slightly attenuated after controlling for adherence to the Mediterranean diet, minutes of MVPA, and school. An inverse association was also found between parental income above the median and the prevalence of obesity and abdominal obesity. This association was not statistically significant for severe obesity (*p* > 0.05). Among the three indicators of SES, income shows the weakest associations with the anthropometric outcomes. The magnitude of these associations was lower in comparison to the other two indicators of SES, educational level, and labor market status. Parental education was the only indicator of SES that showed a significant interaction with adherence to the Mediterranean diet (*p* = 0.024) and MVPA (*p* = 0.016). However, there were no changes in the direction between the exposures and outcomes.

Figure 2 shows the association of the composite SES categories and the prevalence of obesity, severe obesity, and abdominal obesity in children and adolescents participating in this study. The direction and magnitude of the associations evidenced by Model 1 and Model 2 are very similar. According to Model 2, the combination of the most favorable SES indicators (University/Employed/Income above the media) was the least associated with obesity, severe obesity, and abdominal obesity. In particular, in comparison with the combination of the most unfavorable SES indicators (Less than University/Unemployed/Income below the media), the combination of the most favorable SES indicators was associated with a 72%, 80%, and 65% lower likelihood of obesity, severe obesity, and abdominal obesity, respectively. There was no statistically significant interaction between the composite SES categories and children’s and adolescents’ age and gender (*p* > 0.05). 

## 4. Discussion

The aim of this study was to evaluate the association between three parental SES indicators (individually and in combination) and weight status in a nationwide, representative sample of the Spanish child and adolescent population. Parental educational level, labor market status, and annual income per person were inversely related to obesity and abdominal obesity; moreover, parental educational level and labor market status also showed a negative association with severe obesity. Income showed a weaker association with weight status in comparison to educational level and labor market status indicators, but all three SES indicators should be considered when studying pediatric obesity. Finally, children and adolescents living exposed to the most advantageous SES factors were less likely to present obesity, severe obesity, or abdominal obesity than those living exposed to the least favorable SES factors.

Different studies led to opposite conclusions: Miqueleiz et al. reported a statistically significant upward trend in the prevalence of overweightness and obesity between 1987 and 2007 in Spanish girls and boys aged 10 to 15 years [17] but not in 5- to 9-year-old children from families with a lower educational level [17]. Another cross-sectional study with a population aged 2 to 15 years showed a lower prevalence of youth overweightness and obesity within families with a higher SES. This association was reported for all gender and age ranges except for girls under 12 [16]. Data from a longitudinal study that included a large young population from north–east Spain revealed the highest prevalence of overweightness and obesity in all gender and age categories for the most deprived participants [19]. Conversely, another cross-sectional study on the prevalence trend in childhood obesity and SES with a representative sample of a southern Spanish region [20] reported no significant social gradient for the prevalence of excess weight. Although a tendency towards an inverse association was found for the 2011–2012 data, it was less obvious for the 2015–2016 data. The present paper contributes to clarifying the heterogeneous results found by the original studies published so far in Spain. Moreover, it solves previous limitations, such as using anthropometric self-reported data and recruiting regional samples. 

A global systematic review published in 2021 [34] found that in HICs, there was a higher risk for children from disadvantaged backgrounds to have a higher fat mass. Similarly, the present study found an inverse association between parental higher education and being employed with obesity, severe obesity, and abdominal obesity among Spanish children and adolescents. The magnitude of the association was higher for children and adolescents whose parents had a university degree and slightly attenuated by adjusting for adherence to the Mediterranean diet, minutes of MVPA, and school. A recent systematic review identified parental education as the most commonly used SES measure [34] and the one that is more often significantly and inversely associated with childhood overweight/obesity in HICs. The second most commonly used SES indicator is parental income [11]. Our findings are consistent with the above-mentioned ones from European countries. However, in contrast to them, we showed that parental employment status was associated with the prevalence of being overweight [35].

The negative effects of severe childhood obesity on health [36] and its rising prevalence in many HICs are alarming [37]. Although emergent scientific evidence is showing that severe obesity is in general associated with lower socioeconomic status [38], more in-depth analyses are needed to study its association with specific SES indicators [39]. Data from the WHO European Childhood Obesity Surveillance Initiative (COSI) found that in European countries, severe obesity was more common among children whose mothers had a lower educational level. Southern European countries had the highest levels of severe obesity with a prevalence of 4% [40]. The prevalence of severe obesity was 5.1% among children whose mothers had a lower educational level and 2.9% among children whose mothers had a higher educational level [40]. In the present study, a strong and negative association of severe obesity with parental educational level was also obtained. Specifically, children and adolescents whose parents had a university degree were 34% less likely to present severe obesity in comparison with those whose parents had a lower educational level. 

To the best of our knowledge, few studies have reported the association between SES indicators and abdominal obesity in young populations. A cross-sectional study among European adolescents [41] indicates an inverse association, which is consistent with the findings of this study. The magnitude of the association between abdominal obesity and SES indicators reported by Costa de Oliveira Forkert et al. [41] is comparable to the one found in the present study. Moreover, a longitudinal study showed that abdominal obesity was two-folds higher in Spanish children aged 4 to 9 years with parents of low SES in comparison to those with parents of high SES [42]. In our study, children and adolescents exposed to the most favorable SES factors were less likely to present abdominal obesity.

SES inequalities are an important determinant of obesity [11]. The pathways through which inequalities are associated with weight status in the young population are complex and interrelated [43]: structural, community, and individual factors influencing the reaction to food advertising and marketing [44]; food environment [45,46]; access to sports facilities [47] and safe [48] and green spaces [49]; parental role [50,51]; and individual lifestyles [52]. For example, children from socioeconomically disadvantaged backgrounds are disproportionately exposed to unhealthy food advertising [44]. Moreover, those attending schools located in more deprived neighborhoods had fewer opportunities to find establishments selling healthy food products, and they were more exposed to fast food establishments, convenience stores, supermarkets, and grocery stores offering unhealthy foods [45,46]. Additionally, young people in low-SES communities have larger obstacles in participating in sports and physical activities because their neighborhoods might not be safe enough [48]. As for green spaces, in southern European cities, many of the neighborhoods had access to them, but the mean distance was higher for deprived neighborhoods [49]. Additionally, green spaces in the most deprived neighborhoods presented significantly more safety concerns, signs of damage, and a lack of equipment to engage in active leisure activities and had significantly fewer amenities, such as seating or public toilets [49]. Finally, in Madrid, although disadvantaged areas had a shorter distance to the closest sports facility, especially for public and low-cost facilities, the overall number of such facilities was lower in low SES areas [47].

Scientific evidence highlights that families with both low educational levels and income have a greater predisposition to experience psychological distress [53], and children are more likely to be exposed to multiple adversities [54]. Moreover, several studies found that parental stress was associated with childhood obesity [51,55]. Stress and adversity affect parents’ abilities to provide a safe, stable, responsive, and nurturing environment for children [51] and are associated with unhealthy practices, such as fast food consumption by children [55]. Indeed, a higher SES is associated with better health in children, fewer behavioral difficulties, a better quality of life, fewer critical life events, and a healthier lifestyle (less time spent watching television and more practicing physical activity) [52]. In conclusion, children who grow up in circumstances with more socioeconomic risk factors will have a predisposition towards sustained weight growth, whereas those who grow up with more protective factors will better maintain a healthy weight [56]. Lastly, different studies reported that children living in lower socioeconomic backgrounds are more vulnerable to overweightness and obesity later on during childhood or adulthood even if their economic environment improves [57,58].

The main strength of this study is the nationwide, representative design. Furthermore, body weight, height, and waist circumference were measured by trained professionals. Another strength is the use of three different standardized SES indicators to examine the individual and joined association between SES and weight status. Vazquez and Cubbin suggested using multiple measures of SES because one variable cannot capture all aspects embedded in SES [11]. The present analysis includes fewer participants than the total recruited due to missing the different interest variables. However, the cases studied were from all Spanish regions following a very similar percentual distribution than the total invited eligible participants (see Appendix A), which is a strength of this study.

The main limitation of the present study is that the annual mean income per person is based on the census section where the participant schools were located, and using aggregated data is less reliable than using individual data. However, it is known that families commonly choose a school located in their neighborhood [59,60]. Finally, even if some schools are situated at the boundary between two census sections and may enroll students from each of them, we assumed that children and adolescents were living in the same census section of their school [59,60]

Previous scientific evidence highlighted the need to find common indicators to better approximate the SES of an individual [61], and our study represents a step forward in this direction. Future research is needed to harmonize the use of indicators and to enhance comparability between results from different regions and countries [12]. 

The results included in this paper correspond to the first edition of the PASOS project carried out in 2019. Data collection from a second edition will be completed in 2023. It will include a new cross-sectional study and a follow-up of participants from the first edition, allowing for a comparative analysis of the association between SES and weight status over an extended period. Moreover, the longitudinal follow-up of the participants from the first edition will enhance the analysis of the association of obesity, severe obesity, and abdominal obesity with SES. Finally, emerging evidence suggests the need to study the impact of the COVID-19 pandemic on the aforementioned association because a higher increase in weight was observed among young people living in unfavorable socioeconomic conditions in comparison to people living in more favorable socioeconomic conditions [62].

Previous research suggested the need to explore the factors mediating socioeconomic differences in adiposity among children and adolescents to define youth policy development [63]. In a review, Gebremariam et al. reported several factors that could be targeted in interventions aimed at reducing socioeconomic differences in excess weight among young people. These include early life experience (particularly breastfeeding, early weaning, and maternal smoking in pregnancy); child dietary behavior (particularly consumption of sugar-sweetened beverages and breakfast eating patterns); child sedentary activity (particularly watching television and using computers); and maternal BMI [63]. Understanding the mechanisms behind the association of these factors with socioeconomic differences is crucial to tackling childhood obesity by reducing SES inequalities. Moreover, research indicated that programs to prevent childhood obesity tended to reach more people with higher SES levels who often benefit more from innovative initiatives [64,65]. In this regard, it would be interesting to better understand how to target obesity prevention programs for the most disadvantaged population.

A recent report by the Spanish Agency for Food Safety and Nutrition and High Commissioner Against Child Poverty [15] suggests implementing initiatives oriented at facilitating healthy eating in a sustained manner throughout the year at school. This way, it would be possible to address social inequalities; promote daily physical activity and active leisure among school children and reduce sedentary lifestyles; reinforce education and awareness-raising among parents and schoolchildren; and continue to develop epidemiological surveillance of childhood obesity and its determinants, paying particular attention to lowering inequities.

In this sense, the National Strategic Plan for the Reduction of Childhood Obesity (2022–2030) is starting its implementation in Spain [23] and should guide the efforts addressed to prevent childhood obesity overall with a special focus on children and adolescents living exposed to disadvantaged factors. 

## 5. Conclusions

This study demonstrated that Spanish children and adolescents with a lower SES are more likely to present obesity, severe obesity, and abdominal obesity. Therefore, to tackle the obesity epidemic, it is essential to consider the associated social health inequalities.

## Figures and Tables

**Figure 1 nutrients-15-01987-f001:**
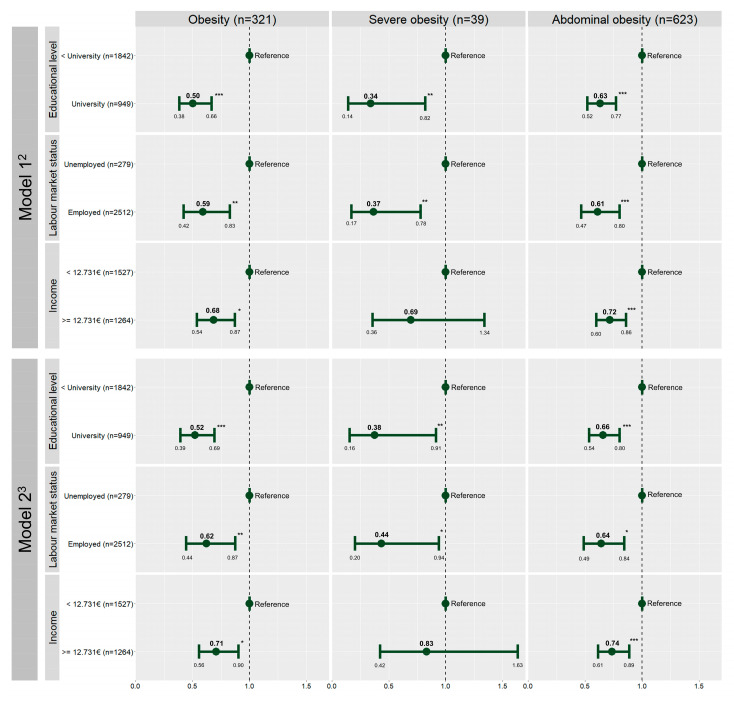
Association between socioeconomic status and general, severe, and abdominal obesity (*n* = 2791)—PASOS study. Spain, 2019–2020. ^1 1^ Performed by logistic regression analysis. ^2^ Model 1: Adjusted for gender and age. ^3^ Model 2: Adjusted for gender, age, adherence to the Mediterranean diet, minutes of MVPA, and school. * *p* < 0.05; ** *p* < 0.01; *** *p* < 0.001.

**Figure 2 nutrients-15-01987-f002:**
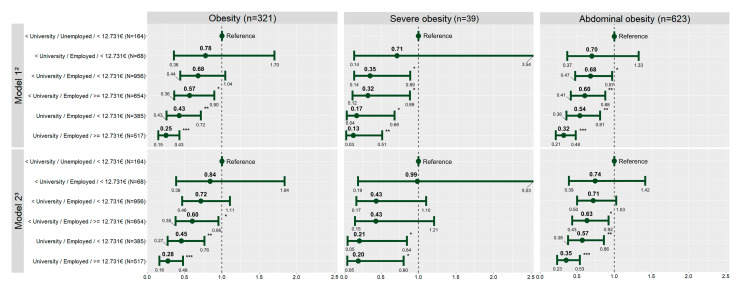
Association between composite socioeconomic status and general, severe, and abdominal obesity (*n* = 2791)—PASOS study. Spain, 2019–2020. ^1 1^ Performed by logistic regression analysis. ^2^ Model 1: Adjusted for gender and age. ^3^ Model 2: Adjusted for gender, age, school, adherence to the Mediterranean diet, and minutes of MVPA. * *p* < 0.05; ** *p* < 0.01; *** *p* < 0.001.

**Table 1 nutrients-15-01987-t001:** Characteristics of the study population (*n* = 2791)–PASOS study. Spain, 2019–2020.

Study Sample Characteristics	Values (*n*; % or Mean)
Girls	1449 (51.9%)
Age	12.59 (12.51–12.68)
BMI (kg/m^2^)	20.29 (20.15–20.44)
Waist (cm)	70.70 (70.30–71.10)
Obesity	321 (11.5%)
Severe obesity	39 (1.4%)
Abdominal obesity	623 (22.3%)
Adherence to the Mediterranean diet (unit)	6.82 (6.73–6.91)
MVPA (min/d)	93.01 (91.75–94.28)
Parental SES indicators	
University degree	949 (34.0%)
Employed	2512 (90.0%)
Annual Income ≥ 12.731€	1264 (45.3%)

Values are expressed as mean (95% CI) and number (%) for continuous variables and proportions, respectively. Body Mass Index (BMI). Moderated/Vigorous Physical Activity (MVPA).

## Data Availability

There are restrictions on the availability of the data for this trial because of the signed consent agreements and concerns regarding data sharing, which only allow access to external researchers for studies following the project purposes. Requestors wishing to access the trial data used in this study can make a request to Santi F. Gómez Santos, PhD (sgomez@gasolfoundation.org) principal researcher of the PASOS study.

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
