# Peer review of "Independent and Joined Association between Socioeconomic Indicators and Pediatric Obesity in Spain: The PASOS Study"

_nutrients, 2023, doi:10.3390/nu15081987_

Round 1

Reviewer 1 Report

The paper addresses an important and interesting topic. However, some wordings are not correct or unclear, and make it difficult to read.   Methods section : The paragraph from L165 to 172 is unclear and needs to be revised.   Results section : - To avoid this distorted aspect of the forest plots (Figure 3), logarithmic scale is often used. - There is some missing information in the titles of figures and tables : number of participants, name of the study, time and place of the study. -What were the effects of MVPA and Mediterranean diet in the models ? Were they significant? Were they as expected ? -Were the interactions between MVPA and socioecoeconomic status and Mediterranean diet and socioeconomic status tested ?   The second part of the conclusion is unclear and needs to be revised.

Comments on the writing :
There are many unclear or incorrect phrasings which need to be clarified: L89, L96, L115-116, L123-124, L144-145, L150, L238, L280, L272, L295, L300-302, L317, L357, L379, L396, L 427, L428-429, L439, L447-451

Author Response

Please see the attachment. Thanks for your comments. 

Reviewer 2 Report

I found reading this manuscript with results from a nationwide representative sample interesting. However, I have made a few comments to improve the manuscript.

Line 94: self-reported anthropometric data instead of anthropometric self-reported data

Line 98: different ??

Lines 101-103: Revise the sentence

Lines 141-143: Deleted, repetition (lines 149-151)

Line 171: data was?

Line 180: 12.73- based on what? Reference?

Line 207: Adjusted P-value for multiple comparisons?

Table 1 : Values, N (%)

Table 2 or Figure?

Merge lines- 302 and 303

Line 467: Assent for children???

References: Consistency (check thoroughly)

Check spelling and grammatical errors thoroughly

Author Response

(The authors gave the same response as above.)
